# Occult Pneumothorax in Blunt Thoracic Trauma: Clinical Characteristics and Results of Delayed Tube Thoracostomy in a Level 1 Trauma Center

**DOI:** 10.3390/jcm12134333

**Published:** 2023-06-28

**Authors:** Chang-Wan Kim, Il-Hwan Park, Young-jin Youn, Chun-Sung Byun

**Affiliations:** Department of Thoracic and Cardiovascular Surgery, Wonju Severance Christian Hospital, Yonsei University Wonju College of Medicine, Wonju 26426, Republic of Korea; asparag@yonsei.ac.kr (C.-W.K.); nicecs@yonsei.ac.kr (I.-H.P.); cution0857@gmail.com (Y.-j.Y.)

**Keywords:** pneumothorax, blunt chest trauma, chest computed tomography, chest X-ray

## Abstract

Occult pneumothorax in blunt trauma patients is often diagnosed only after computed tomography because supine chest X-ray (CXR) is preferred as an initial evaluation. However, improperly managed preexisting occult pneumothorax could threaten the vitality of patients. Therefore, this study aimed to evaluate the incidence, characteristics, risk factors, and outcomes of occult pneumothorax in a single trauma center. From 2020 to 2022, patients who were admitted to the level 1 trauma center were retrospectively investigated. Inclusion criteria focused on blunt chest trauma. Variables including demographic factors, image findings, injury-related factors, tube thoracostomy timing, and treatment results were evaluated. Of the 1621 patients, 187 who met the criteria were enrolled in the study: 32 with overt pneumothorax and 81 with occult pneumothorax. Among all of the pneumothorax cases, the proportion of occult pneumothorax was 71.7% (81/113), and its incidence in all admitted trauma victims was 5.0% (81/1621). Subcutaneous emphysema and rib fractures on supine CXR were risk factors for occult pneumothorax. Six patients underwent delayed tube thoracostomy; however, none had serious complications. Given that occult pneumothorax is common in patients with blunt chest trauma, treatment plans should be established that consider the possibility of pneumothorax. However, the prognosis is generally good, and follow-up is an alternative.

## 1. Introduction

Chest trauma is the third leading cause of trauma-related deaths after abdominal injury and head trauma, accounting for approximately 20–25% of all trauma-related deaths [1,2]. Among blunt chest traumas, traumatic pneumothorax is the second most common injury, following rib fractures, and can be life-threatening if not promptly treated with interventions such as tube thoracostomy [3]. In patients with polytrauma, initial evaluation is often performed using focused assessment with sonography for trauma (FAST) or pelvic immobilization, which may prioritize supine over standing chest X-ray (CXR) [4]. However, supine CXR may be less reliable in detecting traumatic hemothorax or pneumothorax due to overlap with lung shadows compared to standing CXR [5].

Occult pneumothorax has recently been introduced as pneumothorax seen on computed tomography (CT) scans not detected on CXR or in clinical examinations [5]. The importance of CT scanning for diagnosing occult pneumothorax is sometimes overlooked but is crucial in patients with multiple traumas. Early diagnosis based solely on supine CXR is especially important in these patients, as it may involve situations such as the need for endotracheal intubation, difficulty in performing CT scans due to unstable vital signs, or emergency surgery. The occurrence of pneumothorax in such patients and the absence of appropriate treatment can result in fatal consequences for patients with tension pneumothorax or life-threatening conditions [6]. However, it is not widely known how much pneumothorax progress has been made in real situations. Therefore, this study aimed to evaluate the incidence, characteristics, risk factors, and outcomes of occult pneumothorax in patients with blunt chest trauma.

## 2. Materials and Methods

### 2.1. Study Cohort

This study was conducted with patients who visited the level 1 regional trauma center at Wonju Severance Christian Hospital for 2 years, from 1 May 2020 to 1 May 2022. The study focused on patients who met the activation criteria for the trauma team and had thoracic injuries. The criteria for trauma team activation were divided into physiological, anatomical, and mechanism of injury. Detailed criteria are shown in Appendix A. The exclusion criteria for the study were as follows:(1)Patients who died or received cardiopulmonary resuscitation (CPR) within 24 h of admission;(2)Patients transferred to another hospital within 24 h of admission;(3)Patients who did not undergo initial chest CT or CXR;(4)Patients with stabbed chest trauma.

These patients were excluded from the study due to limited data collection and the inability to assess important variables in the study.

### 2.2. Data Collection

Retrospective data including age, sex, mechanisms of injury, revised trauma score (RTS), injury severity score (ISS), tube thoracostomy in the emergency room (ER), positive pressure ventilation within 24 h of admission, delayed tube thoracostomy after admission, length of hospital stay, and mortality were collected. ER tube thoracostomy was defined as chest tube insertion within 1 h of presentation. However, if chest tube insertion was performed after 1 h, it was defined as delayed tube thoracostomy. The presence of pneumothorax, hemothorax, lung contusion, rib fractures, and subcutaneous emphysema was assessed by three thoracic surgeons based on an initial supine CXR. All three thoracic surgeons have a license about the traumatology granted by the Korean Society of Traumatology. During the study period, a single thoracic surgeon was responsible for making initial diagnoses for both chest AP and chest CT during their duty hours. The following day, a discussion involving two additional thoracic surgeons and a radiologist took place to review the radiologic injuries and reach a consensus on final diagnosis. Consequently, the final diagnoses in this study were determined through consensus, and only the final diagnoses were recorded in the electronic medical records. Fractured rib was counted only on CXR. Pneumothorax was classified as occult, which was not visible on the supine CXR but identified on the chest CT, or overt, which was observed on the supine CXR. Occult pneumothorax was further classified as minimal (depth, <1 cm; width, <4 cm), moderate (depth, >1 cm; width, >4 cm and not extending to the mid-coronal line), or large (crossing the mid-coronal line) based on the Wolfman classification [7]. The mid-coronal line was defined as the line dividing the thorax into equal anterior and posterior halves in chest CT. Hemothorax was confirmed by supine CXR as diffuse haziness of the lung field. All chest CT scans were performed using a Philips^®^ iCT 128-slice scanner.

For the ISS, the body is reclassified into six regions (head and neck, face, chest, abdomen and pelvic cavity, pelvis and extremities, and external). The injury severity code is taken for each of the three most severely injured body regions, those three codes are squared, and the three resulting values are added to yield the ISS. Severe trauma is generally defined as an ISS >15. The RTS combines physiological data obtained on patient arrival. The score is calculated based on the Glasgow Coma Scale, systolic blood pressure, and respiratory rate. The RTS ranges from severe (0 points) to normal. Patient data were collected prospectively, and medical records were analyzed retrospectively. This study was approved by the institutional review board of Wonju Severance Christian Hospital, which waived the requirement for informed consent (IRB approval no. CR322100).

### 2.3. Statistical Analysis

All data were stored using Microsoft Excel 2010 (Microsoft, Washington, DC, USA), and all statistical analyses were performed using IBM SPSS Statistics 26 (IBM, New York, NY, USA). One-way analysis of variance and the Kruskal–Wallis test were used for continuous variable analysis, and χ^2^-test and Fisher’s exact test were used for categorical data analysis. Binary logistic regression analysis was employed to assess risk factors. A *p*-value of <0.05 was considered statistically significant.

## 3. Results

During the two-year study period, a total of 1621 patients visited the trauma center. Among them, 267 had chest injuries and were included in the study. However, 80 patients were excluded for different reasons: 24 were transferred to a different institution for primary management; 43 died or had undergone cardiopulmonary resuscitation in ER; 11 did not have initial imaging tests; and two had chest trauma from stabbing. Consequently, 187 patients were ultimately analyzed.

### 3.1. Clinical Characteristics of Patients According to Occult Pneumothorax

There were 74 (39.6%) and 113 patients (60.4%) without and with pneumothorax, respectively. Among the patients with pneumothorax, 81 (43.3%) had occult pneumothorax and 32 (17.1%) had overt pneumothorax. The rate of occult pneumothorax among all pneumothorax cases was 71.7% (81/113), and the prevalence of occult pneumothorax compared to all hospitalized patients was 5.0% (81/1621). After classifying the study patients into three groups (without pneumothorax, occult pneumothorax, and overt pneumothorax), several variables were analyzed for statistically significant differences in the three groups. The results showed that the RTS was significantly lower in the overt pneumothorax group than in the other groups (*p* = 0.001). Although not statistically significant, there was an increase in the number of positive pressure ventilations and the lengths of stay in overt pneumothorax. Appendix A displays relevant patient data for the study groups. Appendix A presents the analysis of injuries observed on supine CXR. Subcutaneous emphysema more commonly occurred in the overt pneumothorax group (*p* = 0.000), whereas the presence (*p* = 0.007) and number (*p* = 0.009) of rib fractures were greater in the occult pneumothorax group. As shown in Appendix A, the mean tube thoracostomy time was 109.1 ± 596.8 min. The mean tube thoracostomy time for occult pneumothorax was 243.7 ± 892.0 min.

Of the 81 patients with occult cases, 36 had minimal (44.4%), 31 had moderate (38.3%), and 14 had large (17.3%) pneumothoraxes. In these groups, ISS increased with pneumothorax size, and the number of ER tube thoracostomies and intubations for positive pressure ventilation also increased. The minimal occult pneumothorax group had the shortest hospitalization period. Data for this patient group are presented in Table 1. When examining their CXR injury findings, lung contusion (*p* = 0.015) significantly increased with larger occult pneumothoraxes, and the presence and number of rib fractures (*p* = 0.013) also increased. Subcutaneous emphysema (*p* = 0.003) was statistically most common in the moderate occult pneumothorax group. Data for this patient group are presented in Table 2.

### 3.2. Risk Factors of Occult Pneumothorax in Blunt Chest Trauma

An analysis was conducted to identify risk factors associated with occult pneumothorax. Variables such as age, sex, and supine CXR findings including lung contusion, hemothorax, rib fracture, and subcutaneous emphysema were examined for their association in the groups without pneumothorax and with occult pneumothorax. Table 3 presents significant variables of occult pneumothorax identified through univariate analysis, while Table 4 shows the results of multivariate analysis. The results show that the presence of rib fracture (*p* = 0.003; odds ratio [OR], 3.270; 95% confidence interval [CI], 1.476–7.243) and subcutaneous emphysema (*p* = 0.031; OR, 4.278; 95% CI, 1.138–16.087) on supine CXR were significant risk factors for occult pneumothorax.

### 3.3. Delayed Tube Thoracostomy after Positive Pressure Ventilation

In this study, we aimed to evaluate the clinical outcomes of delayed tube thoracostomy in patients who received positive pressure ventilation and did not undergo tube thoracostomy in the ER. To achieve this, 42 patients with occult pneumothorax who underwent positive pressure ventilation were analyzed, as shown in Table 5. Overall, delayed tube thoracostomy was observed in 14.3% (6/42) of the occult pneumothorax patients. Among these, 16.7% (1/6) of patients with minimal occult pneumothorax who underwent positive pressure ventilation received delayed tube thoracostomy, while none of the patients with moderate occult pneumothorax who underwent positive pressure ventilation received delayed tube thoracostomy. Furthermore, among the patients who did not undergo positive pressure ventilation, 15.4% (4/26) of those with minimal occult pneumothorax and 14.3% (1/7) of those with moderate occult pneumothorax received delayed tube thoracostomy. No statistically significant differences were observed in any of the above results. The mean delayed tube thoracostomy time was 52.0 ± 23.6 h.

## 4. Discussion

### 4.1. Incidence of Occult Pneumothorax

Pneumothorax is a condition in which air accumulates in the chest cavity. In a standing CXR, air tends to accumulate at the lung apex due to its lower density compared to the lung tissue, pleural fluid, or blood in the chest cavity, which can result in the appearance of pneumothorax at the top portion of the lung. However, in a supine CXR, most of the pneumothorax may overlap with the lung tissue, making it difficult to diagnose, and this type of pneumothorax has been reported to range from 30% to 68% of all patients with pneumothorax [3,8]. In this study, 81 patients had occult pneumothorax, accounting for 71.7% of the 113 patients with pneumothorax and 5% of the 1621 patients visiting the trauma center. The incidence of occult pneumothorax was higher in this study compared to recent studies, which may be because previous studies primarily relied on abdominal CT scans to confirm occult pneumothorax, whereas chest CT scans were performed for all patients as protocol at the level 1 trauma center in this study, leading to a higher diagnostic rate. The term “occult pneumothorax” was initially used to describe pneumothorax that cannot be detected on regular X-rays but can be detected on abdominal or cervical CT scans, and its definition has gradually expanded to include pneumothorax detected on chest CT scans [5,9]. Recent studies have reported that approximately 5% of all hospitalized patients exhibit an incidental pneumothorax [9].

### 4.2. Risk Factors of Occult Pneumothorax

Previous studies have reported that subcutaneous emphysema and rib fractures are commonly identified as risk factors for occult pneumothorax [5,9]. This study also identified statistically significant risk factors associated with subcutaneous emphysema and rib fractures on CXR in the occult pneumothorax group, as shown in Figure 1. Recently, FAST has become a widely used diagnostic tool for pneumothorax in patients with trauma, with reported sensitivity rates ranging from 92% to 100% [4]. However, the presence of subcutaneous emphysema in ultrasound can limit the field of view, which can make it challenging to accurately diagnose pneumothorax using FAST in patients with trauma. Paradoxically, subcutaneous emphysema is also an important predictor of pneumothorax [5].

Rib fractures are often difficult to accurately diagnose on CXR. In general, chest CT and bone scans are used to accurately diagnose rib fractures. Chapman et al. reported that approximately 75% of rib fractures are not visible on CXR, and about half of diagnosed rib fractures are confirmed by CT findings [10]. However, this study used CXR-visible rib fractures as our criteria, as rib fractures that are visible only with CT have no predictive value for occult pneumothorax. Rib fractures identified on CT scans may not have significant clinical implications, as CT can also detect minor displacements. It is clinically more valuable to examine rib fractures that are clearly visible on CXR. Furthermore, this study found that the presence of rib fractures on CXR was higher in patients with occult pneumothorax than in those with overt cases or without pneumothorax. In addition, the number of rib fractures was higher in patients with occult pneumothorax than in those with overt pneumothorax, despite the small difference in the average number of rib fractures. This could be due to the exclusion of patients who died or underwent CPR, who are presumed to have had more rib fractures due to more severe injuries. As a result, the statistically significant difference is limited in assigning high clinical value to the number of rib fractures detected on CXR as a predictor of occult pneumothorax. Therefore, if pneumothorax is not visible on CXR but a rib fracture is detected, close examination for subcutaneous emphysema is warranted.

### 4.3. Progression to Delayed Pneumothorax during Hospitalization

The treatment approach for occult pneumothorax remains controversial. While tube thoracostomy is recommended for traumatic hemopneumothorax, the classification or scoring system for determining the severity or extent of hemopneumothorax is yet to be elucidated. In addition to the Wolfman classification used in this study, de Moya et al. [11] introduced a scoring system based on the presence or absence of depth of invasion and hilar height. Ball et al. [12] proposed a calculation method that involves multiplying the maximum width of pneumothorax by the number of images showing pneumothorax on chest CT; however, no widely accepted classification or scoring system is currently available. Despite several studies advocating the conservative management of occult pneumothorax, many clinicians prefer invasive methods, such as chest tube insertion [5,6,7,8,9,11,12,13,14,15,16]. In particular, when endotracheal intubation is required, tube thoracostomy is commonly performed preemptively due to concerns about worsening pneumothorax. Not surprisingly, all large occult pneumothorax in this study underwent early tube thoracostomy according to the trauma physician’s preference.

The association between positive pressure ventilation and delayed tube thoracostomy is considered an important factor in predicting the prognosis of patients with occult pneumothorax during hospitalization. Among the 42 patients observed for occult pneumothorax in this study, delayed tube thoracostomy was performed in only six, resulting in a rate of 14.3%. In a study by Hershkovitz et al. [17], although there was no statistical difference, 21.1% of patients who underwent positive pressure ventilation received delayed tube thoracostomy, compared to 13.6% of those who did not undergo positive pressure ventilation. The authors concluded that most patients with occult pneumothorax who undergo positive pressure ventilation would be safe, but a small number may still require delayed tube thoracostomy. In our study, nine patients received positive pressure ventilation within 24 h of hospitalization, but only one patient (11.1%) received delayed tube thoracostomy. None of the patients who received intubation for positive pressure ventilation experienced an increase in pneumothorax. These results also support the conclusion that delayed pneumothorax may not always be increased by positive pressure ventilation. Therefore, routine tube thoracotomy may not be necessary based on these findings, and in particular, delayed tube thoracostomy may not be necessary for all patients with occult pneumothorax who receive positive pressure ventilation and do not undergo tube thoracostomy in the ER.

### 4.4. Management of Occult Pneumothorax

This study has reported no statistically significant difference between patients with occult pneumothorax who undergo tube thoracostomy and those who do not. Although statistical significance may not have been achieved due to the small sample size, simple follow-up observation may be sufficient for small traumatic pneumothorax. Furthermore, previous studies have shown that conservative treatment without tube thoracostomy may yield favorable results in patients with small pneumothorax, even in those undergoing mechanical positive pressure ventilation [3,5,7,9,12,13,14,15,16]. Some studies have shown that complications related to tube thoracostomy occur in approximately 25% of patients with trauma, leading to prolonged hospitalization periods [16,18,19]. As a result, closely monitoring the progression of pneumothorax could also be a viable approach. In this study, our findings suggest that delayed tube thoracostomy may not be necessary for all patients who receive positive pressure ventilation and do not undergo immediate tube thoracostomy. However, further studies are needed to develop an improved classification for occult pneumothorax and appropriate treatment strategies.

This study has some limitations that should be acknowledged. First, the findings may not be generalizable to other trauma centers as the study only represents the experience of a single trauma center. Second, the study period coincided with the COVID-19 pandemic, which had a significant impact on the epidemiology of trauma patients and the overall operation of hospitals nationwide. Therefore, it is reasonable that it may have influenced the results of study. However, it is important to note that the operation of level 1 regional trauma centers in South Korea experienced minimal changes. These centers receive separate government financial support, have dedicated human resources, and undergo continuous government supervision and audits. Third, the study was conducted retrospectively, relying on electronic medical records and picture archiving and communication systems, which may have limitations in accuracy and completeness. Fourth, the kappa coefficient for the three thoracic surgeons and one radiologist was not evaluated because only final radiologic diagnoses were recorded in the electronic medical records through consensus. Moreover, the study only included patients with blunt chest trauma via inclusion and exclusion criteria, and the results therefore may not be indicative of critically ill patients who underwent CPR or were transferred via other hospitals. It is important to note that the decision to insert a chest tube should not be solely dependent on the attending trauma physician.

## 5. Conclusions

Occult pneumothorax was found in approximately 5% of all patients with trauma and 70% of all patients with traumatic pneumothorax. Therefore, when surveying patients with trauma, the possibility of occult pneumothorax being not visible on the initial supine CXR should be considered, and significant findings and treatment plans should be examined accordingly. Conservative management with careful observation should be considered before deciding on tube thoracostomy in patients with minimal pneumothorax volume. In addition, further research on pneumothorax volume, associated injuries, and appropriate treatment strategies should be conducted.

## Figures and Tables

**Figure 1 jcm-12-04333-f001:**
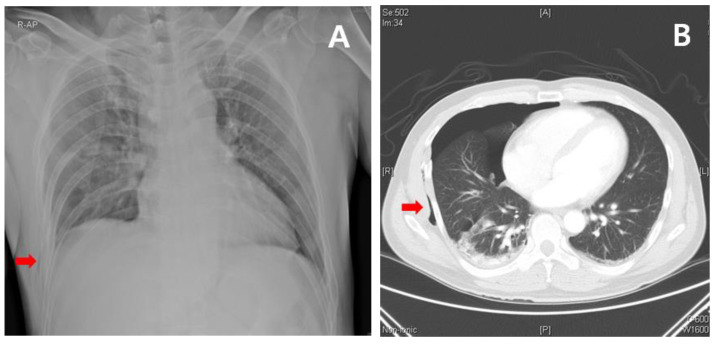
Typical finding of occult pneumothorax. (**A**) Subcutaneous emphysema (arrow) and rib fracture are observed on supine chest X-ray. (**B**) Chest computed tomography reveals subcutaneous emphysema (arrow), rib fracture, and occult pneumothorax.

**Table 1 jcm-12-04333-t001:** Comparison of characteristics according to size of occult pneumothorax.

	Total (*n* = 81)	Minimal Occult Pneumothorax (*n* = 36)	Moderate Occult Pneumothorax (*n* = 31)	Large Occult Pneumothorax (*n* = 14)	*p*-Value
Age	54.23 ± 16.27	52.81 ± 16.45	56.61 ± 18.60	52.64 ± 16.27	0.590
Male	65 (80.2)	31 (86.1)	22 (71.2)	12 (85.7)	0.279
Injury mechanism					N/A
Falls	29 (35.8)	12 (33.3)	12 (38.7)	5 (35.7)	
Car accident	23 (28.4)	10 (27.8)	8 (25.8)	5 (35.7)	
Pedestrian accident	15 (18.5)	5 (13.9)	8 (25.8)	2 (14.3)	
Motor/bicycle	11 (13.6)	8 (22.2)	3 (9.7)	0 (0.0)	
Collision	3 (3.7)	1 (2.8)	0 (0.0)	2 (14.3)	
RTS	7.66 ± 0.68	7.79 ± 0.22	7.71 ± 0.39	7.24 ± 1.45	0.240
ISS	18.37 ± 7.80	15.94 ± 7.07	17.65 ± 7.29	26.21 ± 5.75	0.000 *
ER tube thoracostomy	39 (48.1)	4 (11.1)	21 (67.7)	14 (100.0)	0.000 *
Positive pressure ventilation	32 (39.5)	9 (25.0)	14 (45.2)	9 (64.3)	0.029 *
Length of stay	31.68 ± 39.02	20.06 ± 18.11	41.06 ± 57.30	40.79 ± 15.32	0.002 *
Mortality	1 (1.2)	0 (0.0)	1 (3.2)	0 (0.0)	0.556

Values are presented as mean ± standard deviation or number (%). * *p* < 0.05. Abbreviations: ER-emergency room; ISS-Injury Severity Score; RTS-Revised Trauma Score.

**Table 2 jcm-12-04333-t002:** Injury type by chest X-ray according to size of occult pneumothorax.

	Total (*n* = 81)	Minimal Occult Pneumothorax (*n* = 36)	Moderate Occult Pneumothorax (*n* = 31)	Large Occult Pneumothorax (*n* = 14)	*p*-Value
Lung contusion	15 (18.5)	3 (8.3)	6 (19.4)	6 (42.9)	0.015 *
Hemothorax	17 (21.0)	5 (13.9)	7 (22.6)	5 (35.7)	0.221
Subcutaneous emphysema	13 (16.0)	1 (2.8)	10 (32.3)	2 (14.3)	0.003 *
Rib fracture	30 (37.0)	8 (22.2)	12 (38.7)	10 (71.4)	0.004 *
Number of fractured ribs	1.06 ± 1.59	0.64 ± 1.38	1.19 ± 1.78	1.79 ± 1.42	0.015 *

Values are presented as mean ± standard deviation or number (%). * *p* < 0.05.

**Table 3 jcm-12-04333-t003:** Univariate analysis for risk factors of occult pneumothorax.

	No Pneumothorax (*n* = 74)	Occult Pneumothorax (*n* = 81)	*p*-Value
Age	55.01 ± 20.40	54.23 ± 16.27	0.794
Male	51	65	0.138
RTS	7.56 ± 0.96	7.66 ± 0.68	0.439
ISS	20.27 ± 8.36	18.37 ± 7.78	0.147
Injury type of chest X-ray			
Lung contusion	11	15	0.668
Hemothorax	13	17	0.685
Rib fracture	11	30	0.002 *
Subcutaneous emphysema	3	13	0.017 *

Values are presented as mean ± standard deviation or number (%). * *p* < 0.05. Abbreviations: ISS-Injury Severity Score; RTS-Revised Trauma Score.

**Table 4 jcm-12-04333-t004:** Multivariate analysis for risk factors of occult pneumothorax.

	Odds Ratio (95% CI)	*p*-Value
Rib fracture	3.270 (1.476–7.243)	0.003 *
Subcutaneous emphysema	4.278 (1.138–16.087)	0.031 *

* *p* < 0.05. Abbreviations: CI-confidence interval.

**Table 5 jcm-12-04333-t005:** Relationship between positive pressure ventilation and delayed tube thoracostomy.

		Positive Pressure Ventilation			
		(*n* = 33)	(*n* = 9)		
	Delayed Tube Thoracostomy	−	+	*p*-Value	Overall *p*-Value
Minimal occult pneumothorax (*n* = 32)	−	22	5	NS	NS
	+	4	1		
Moderate occult pneumothorax (*n* = 10)	−	6	3	NS	
	+	1	0		

Abbreviations: NS-not significant.

## Data Availability

The data that support the findings of this study are available from the corresponding author upon reasonable request.

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
