# Peer review of "Occult Pneumothorax in Blunt Thoracic Trauma: Clinical Characteristics and Results of Delayed Tube Thoracostomy in a Level 1 Trauma Center"

_jcm, 2023, doi:10.3390/jcm12134333_

Round 1

Reviewer 1 Report

*  In general, please refrain from the use of manufactured acronyms (such as OPX, PPV, DTT) which generally only serve to frustrate the reader.  Plese spell them out instead (we have plenty of electron space).  Commonly used acronyms, such as CXR, RTS/ISS, ER, and FAST are acceptable.

*  Why was the data analysis limited to only 2 years?  My guess is that your centre has such data going back several years.  Additionally, this time period was an outlier in regards to hospital operations related to the COVID epidemic.  Does this analysis apply to more 'normal times'?  This can be answered by expanding the period of analysis using the same techniques.

*   For the statistics, what was the agreement among the three surgeons for the CXR assessment?  I'm guessing it was quite high, and including this analysis will strengthen the conclusions.  BTW, was hemothorax in CXR assessed in this way too?

*  In the Results, was the larger group with pneumothorax only those seen on CXR?  This is clearer in the next sentence, but it would read better if the phrase (overt and occult).  

*  What is missing from the analysis is the time from initial diagnosis of occult pneumothorax to chest tube insertion in those undergoing this treatment.  Was this in hours or days?  This should be made quite clear, and is only mentioned peripherally in the Discussion.  What signs/symptoms developed inducing placement of a chest tube?  Over what time course?  

Author Response

*  In general, please refrain from the use of manufactured acronyms (such as OPX, PPV, DTT) which generally only serve to frustrate the reader.  Plese spell them out instead (we have plenty of electron space).  Commonly used acronyms, such as CXR, RTS/ISS, ER, and FAST are acceptable.

As the reviewer said, the abbreviation has been corrected. thank you.

*  Why was the data analysis limited to only 2 years?  My guess is that your centre has such data going back several years.  Additionally, this time period was an outlier in regards to hospital operations related to the COVID epidemic.  Does this analysis apply to more 'normal times'?  This can be answered by expanding the period of analysis using the same techniques.

Thank you for Reviewer's Comment.

1621 patients were inevitably limited to two years because three thoracic surgeons (also trauma specialists) reviewed all CXR, chest CT and electric medical records retrospecvely at the time of visit and after hospitalization. It would be nice to extend the analysis period further, but we will research with more resources for fancy study designs in the future.

And although you concerned our research time, we cannot agree that Covid has changed the epidemic or treatment method of trauma patients. In South Korea, there are statistics showing that wearing a mask and penetrating injuries decreased during Covid, but there was no significant change in blunt trauma.

*   For the statistics, what was the agreement among the three surgeons for the CXR assessment?  I'm guessing it was quite high, and including this analysis will strengthen the conclusions.  BTW, was hemothorax in CXR assessed in this way too?

Thank you for your comments.

Three of the authors are thoracic surgeons with licensed for Traumatology. One of them reviewed trauma patients who visited the trauma center on every duty day. The following day, at least two other thoracic surgeons and one radiologist reviewed chest radiography and chest CT for ISS documentation and confirmed the diagnosis.

I added this to the method. All diagnoses including Hemothorax were evaluated in the same way. However, We diagnosed differently lung contusion and hemothorax via radiologic finding. Hazziness of lung contusion is not restricted by the anatomical boundaries of the lobes, or segments of the lung. And geographic, non-segmental areas of ground-glass densities were typical. However, hemothorax is diffuse haziness on entire hemithorax on CXR.

*  In the Results, was the larger group with pneumothorax only those seen on CXR?  This is clearer in the next sentence, but it would read better if the phrase (overt and occult).  

Thank you for your comment.

In the results, all lager pneumothorax are the contents expressing the amount of occult pneumothorax. This is clearly stated in the paper as occult.

*  What is missing from the analysis is the time from initial diagnosis of occult pneumothorax to chest tube insertion in those undergoing this treatment.  Was this in hours or days?  This should be made quite clear, and is only mentioned peripherally in the Discussion.  What signs/symptoms developed inducing placement of a chest tube?  Over what time course?  

Thank you for your comment.

ER tube thoracostomy was defined chest tube insertion within 1 hour after visiting. However, if chest tube insertion was performed after 1 hours, it was defined as delayed tube thoracostomy. We added this part to method. Although data were not shown in the paper, the average time for delayed tube thoracostomy was approximately 12.32 hours. However, it was not shown because the number was small, and the criteria and time of tube insertion were not performed according to a certain standard.

And chest tube insertion was performed if an overt pneumothorax was visible, regardless of symptoms. However, even with a small pneumothorax (occult pneumothorax in this study), if repiratory distress or chest wall pain were complained, chest tube insertion was performed according to the judgment of an on duty thoracic surgeon or emergency physician.

Reviewer 2 Report

Thank you very much for the opportunity to review a scientific article on pneumothorax after blunt chest trauma.

The article is very well written, it contains all necessary sections. The quality of the English language is very good. I have no major comments on the article.

The authors briefly mention the use of ultrasonography in the diagnosis of pneumothorax. I believe this part could be expanded a bit in the discussion, especially in the aspects of effectiveness of diagnosis, limitations and possible use to follow-up the pneumothorax.

References to references should be in the form "[4]." instead of ".[4]". Please make the appropriate corrections.

Other than that, I have no other comments. Once again, congratulations to the authors of the articles

Quality of English language is high.

Author Response

Thank you very much for the opportunity to review a scientific article on pneumothorax after blunt chest trauma.

The article is very well written, it contains all necessary sections. The quality of the English language is very good. I have no major comments on the article.

The authors briefly mention the use of ultrasonography in the diagnosis of pneumothorax. I believe this part could be expanded a bit in the discussion, especially in the aspects of effectiveness of diagnosis, limitations and possible use to follow-up the pneumothorax.

References to references should be in the form "[4]." instead of ".[4]". Please make the appropriate corrections.

Other than that, I have no other comments. Once again, congratulations to the authors of the articles

Thank you so much for the Reviewer's comment.

I have corrected what you pointed out.

Round 2

Reviewer 1 Report

*  For the measurement period, the excuse of the authors having to do more work to confirm the findings is probably not the best answer.  Worldwide, the COVID epidemic changed many things in hospital operations which may or may not be evident, even if the number of patients was the same.  To confirm your findings, it should be equivalent during a time that this was not an issue.  Including just one year, say 2019, with no differences in the analysis would prove to any reader who is skeptical about the results because of this world-wide event.  I highly recommend that this be done to settle the issue.

*  For the agreement among the surgeons, this is generally done via a kappa statistic as a measurement of agreement.  This statistic or something similar should be added to convince the reader of the results.

*  Lastly, my final comment about time to chest tube placement should be included in the paper, even if the numbers are small.  Again, the skeptical reader will be more convinced if this information is forthcoming.

 Minor editing of English language required

Author Response

*  For the measurement period, the excuse of the authors having to do more work to confirm the findings is probably not the best answer.  Worldwide, the COVID epidemic changed many things in hospital operations which may or may not be evident, even if the number of patients was the same.  To confirm your findings, it should be equivalent during a time that this was not an issue.  Including just one year, say 2019, with no differences in the analysis would prove to any reader who is skeptical about the results because of this world-wide event.  I highly recommend that this be done to settle the issue.

Answer)

I agree with what the reviewer said. According to 'The Epidemiology of Major Trauma During the First Wave of COVID-19 Movement Restriction Policies: A Systematic Review and Meta-analysis of Observational Studies,' published in 2022 by Marcello Antonini and others, as well as recent papers published in worldwide, It is clear that during the COVID-19 period, the distribution of trauma patients changed globally. Italian researchers found that the number of trauma patients decreased by about 25-30%, and research published in England also confirmed a 21% decrease in the total number of trauma patients. And this result may changed the hospital operation.

Additionally, in South Korea, the National Emergency Medical Center (NEMC) published the annual statistical trauma report, which includes the number of patients visiting 17 trauma centers in the country. You can find the data at the following link:

(https://www.e-gen.or.kr/nemc/statistics_annual_report.do?brdclscd=04) last accessed 18-06-2023.

Here are the National database statistics for the years mentioned:

Pre-Covid-19 period.

2018: All trauma center visits - 37,372; severe trauma (ISS≥15) - 8,299 (22.2%)

2019: All trauma center visits - 37,635; severe trauma (ISS≥15) - 8,892 (23.6%)

Covid-19 period.

2020: All trauma center visits - 34,318; severe trauma (ISS≥15) - 8,918 (26.0%)

2021: All trauma center visits - 34,835; severe trauma (ISS≥15) - 8,906 (25.6%)

Looking at the number of patients visiting the 17 trauma centers before and after the COVID-19 period, it can be observed that from 2020, the total number of trauma patients decreased by 7.8%, the number of severe trauma patients increased by 3.7%, and the ratio increased by 12.7%.

We acknowledge that there were significant changes in most hospital operations during the COVID-19 period. However, it is worth noting that the 17 regional trauma centers in South Korea experienced minimal changes. In our hospital, additional infection control facilities were implemented, and a 24-hour infection control system was established. On the other hand, the regional trauma centers, being institutions subject to annual government management and quality control audits, continued to operate independently without any fluctuations from the hospital administration.

We apologize for not specifically conducting this study throughout the entire COVID-19 period. Unfortunately, due to time constraints during the revision process, we were unable to further validate the results by adding an additional year of data. However, the limitations of the study will mention the changed by the COVID-19 period. We sincerely appreciate the valuable feedback from the reviewer.

*  For the agreement among the surgeons, this is generally done via a kappa statistic as a measurement of agreement.  This statistic or something similar should be added to convince the reader of the results.

Answer)

The Kappa coefficient values for each positive diagnosis between chest AP and chest CT were evaluated by a statistics professor, as indicated in the figure below, in additional response to the reviewer's comment.

Ratio of positive findings on Chest AP when Chest CT scan in positive in blunt trauma patients.

Accuracy(%)

Sensitivity(%)

Specificity(%)

F-score(%)

Kappa
Coefficient

Rib fracture

74.9%

72.8%

100.0%

84.3%

0.292

Lung contusion

50.5%

43.8%

99.4%

60.9%

0.155

Subcutaneous . emphysema

83.5%

56.0%

99.9%

71.7%

0.613

pneumothorax

46.7%

12.7%

100.0%

22.6%

0.102

However, we cannot calculate Kappa coefficients for each physician some critical fault. During the study period, a single thoracic surgeon was responsible for making initial diagnoses for both chest AP and chest CT during their duty hours. The following day, a discussion involving two additional thoracic surgeons and a radiologist took place to review the radiologic injuries and reach a consensus agreement on the final diagnoses. Since the final diagnoses in this study were determined through consensus, and only the final diagnosis was recorded in the Electronic Medical Records, it was not possible to review individual physicians' diagnoses or calculate Kappa coefficients for each physician.

We acknowledge this limitation and have provided additional explanation in the methodology section to address it. We appreciate the valuable feedback provided by the reviewer.

*  Lastly, my final comment about time to chest tube placement should be included in the paper, even if the numbers are small.  Again, the skeptical reader will be more convinced if this information is forthcoming.

Answer)

Chest tube insertion time was measured as 20.5+7.7 minutes for Overt pneumothorax, 243.7+892.0 minutes for Occult pneumothorax, and 52.0 ± 23.6 hours for delayed tube thoracostomy. This chest tube insertion time was added to Result and the supplement table 1. Thanks for the reviewer's comment.

Minor editing of English language was done.
